# Isolation of Intestinal Macrophage Subpopulations for High-Quality Total RNA Purification in Zebrafish

**DOI:** 10.3390/mps7030043

**Published:** 2024-05-17

**Authors:** Yalén Del Río-Jay, Audrey Barthelaix, Cristian Reyes-Martínez, Christophe Duperray, Camila J. Solis-Cascante, Yessia Hidalgo, Patricia Luz-Crawford, Farida Djouad, Carmen G. Feijoo

**Affiliations:** 1Fish Immunology Laboratory, Facultad Ciencias de la Vida, Universidad Andrés Bello, Avenida República 330, Santiago 8370186, Chile; 2Institute for Regenerative Medicine & Biotherapy (IRMB), Centre Hospitalier Universitaire de Montpellier Hôpital Saint Eloi, University Montpellier, INSERM, 80 Avenue Augustin Fliche, 34295 Montpellier, France; 3Montpellier Ressources Imagerie (MRI), BioCampus, University of Montpellier, CNRS, INSERM, 34295 Montpellier, France; 4IMPACT, Center of Interventional Medicine for Precision and Advanced Cellular Therapy, Avenida Plaza 2501, Santiago 7620157, Chile; 5Laboratory of Nano-Regenerative Medicine, Faculty of Medicine, Universidad de los Andes, Las Condes, Santiago 7620157, Chile; 6Laboratorio de Inmunología Celular y Molecular, Facultad de Medicina, Centro de Investigación Biomédica, Universidad de los Andes, Avenida Monseñor Álvaro del Portillo 12455, Santiago 7620157, Chile

**Keywords:** zebrafish, intestinal macrophage, mRNA

## Abstract

Intestinal macrophages have been poorly studied in fish, mainly due to the lack of specific molecular markers for their identification and isolation. To address this gap, using the zebrafish *Tg*(*mpeg1*:EGFP) transgenic line, we developed a fluorescence-activated cell sorting strategy (FACS) that allows us to isolate different intestinal macrophage subpopulations, based on GFP expression and morphological differences. Also, we achieved the purification of high-quality total RNA from each population to perform transcriptomic analysis. The complete strategy comprises three steps, including intestine dissection and tissue dissociation, the isolation of each intestinal macrophage population via FACS, and the extraction of total RNA. To be able to characterize molecularly different macrophage subpopulations and link them to their functional properties will allow us to unravel intestinal macrophage biology.

## 1. Introduction

Intestinal macrophages are specialized myeloid cells intricately involved in orchestrating both homeostasis and inflammatory responses within the gut. Unlike mammals, our understanding of macrophage diversity in other vertebrates, such as fish, remains limited. This knowledge gap primarily stems from the scarcity of information regarding the distinct molecular markers expressed in fish macrophages, which could be employed for precise identification and isolation. These markers are essential for subsequent investigations, including transcriptional profiling and characterization. Thus, an analysis of simpler parameters, such as differences in morphology, is an alternative for identifying different macrophage populations. In this context, we hypothesize that zebrafish larvae are an ideal model for the four-dimensional exploration of the diversity of macrophage subpopulations based on their ability to adopt different morphologies depending on their microenvironments. To accomplish this goal, we employed *Tg*(*mpeg1*:EGFP) transgenic fish, where macrophages are fluorescently marked, and subjected them to an inflammatory diet to trigger intestinal inflammation [1]. Under this condition, we were able to discern two macrophage subpopulations based on distinct morphological characteristics within the intestine: one exhibited a rounded shape, while the other displayed a stellate morphology [2]. To isolate each population, we employed fluorescence-activated cell sorting (FACS) to identify GFP^+^ cells, segregating them into two groups based on their fluorescence signal and forward scatter (FSC) parameter: small GFP^+^ cells representing the round-shaped cells, and large GFP^+^ cells corresponding to the stellate-shaped cells. Subsequently, we conducted a comparative analysis of mRNA expression for specific immune gene markers through RT-qPCR. The comprehensive strategy encompasses the following sequential steps: (1) the dissection of the intestine and subsequent cell dissociation; (2) the isolation of intestinal macrophages, followed by separating each subpopulation via FACS; (3) the extraction of total RNA and the subsequent analysis of gene expression using RT-qPCR.

The main challenges encountered during this process were as follows: (1) the small size of zebrafish macrophages posed a substantial complexity in their isolation, especially amidst the presence of cellular debris and in distinguishing between the two distinct macrophage populations; (2) the limited quantity of GFP^+^ cells obtained after sorting in each subpopulation resulted in less efficient total RNA extraction. Consequently, genes with low expression levels were inadequately represented.

Through this protocol, we present a solution for the efficient separation of both GFP^+^ populations and the achievement of high-quality total RNA extraction, addressing these challenges and facilitating robust gene expression analyses.

## 2. Experimental Design

### 2.1. Protocol Specifications

#### Zebrafish Lines Maintenance

Fish maintenance and husbandry were carried out within the fish facility at the University of Montpellier, adhering to well-established protocols [3]. Adult zebrafish were housed in 3.5 L polycarbonate tanks integrated into a recirculation system, maintaining a constant environment with a water temperature of 28 °C, the conductivity set at 500 μS, a pH of 7.0, and was subjected to a daily light cycle of 14 h, followed by 10 h of darkness. Feeding regimes included two daily servings of dry food (Skreting GEMMA Micro 500, France) along with one feeding of *Artemia nauplii* (Sera, Artemia-mix). All experiments were performed with either AB wild-type fish or and *Tg*(*mpeg1*:EGFP) transgenic fish. In the latter, the expression of the GFP is driven by the *mpeg1* promoter specifically to macrophages, as outlined by Ellett et al. [4].


**Protocol Details**


(1)
*
**The Dissection and Disaggregation of the Intestine from Adults AB Wild-Type and Tg(mpeg1:EGFP) zebrafish.**
*

**Materials**
AB wild-type and *Tg*(*mpeg1*:EGFP) adult zebrafish (either one female or one male of approximately 12 months old) can be chosen indistinctly by line.Petri dishes, diameter: 94 mm; height: 16 mm (Cat# 633181)Sterile scalpelSterile dissecting forceps (Dumont #5; Cat# NC9889584)Ice coolerSterile plastic tubes, 50 mL (Cat# 4610-1943)Cell Strainer 70 μm mesh (Cat# 352350)Cell Strainer 40 μm mesh (Cat# 352340)Cell Strainer 20 μm mesh (Cat# 04-004-2325)P1000 pipettes and tipsPlastic graduated Pasteur pipettes (Gauss, Cat# PIPP-003-500)Microfuge tubes, 1.5 mL (Cat# CFT-001-015)Round Bottom Polystyrene FACS tube, 5 mL (Falcon, Cat#352052)
**Equipment**
Stereomicroscope (Cat. NZ.1903-S)Cooling centrifuge
**Reagents**
Ethanol, 70%Sterile distilled waterDulbecco’s Phosphate-Buffered Saline (DPBS), 1X, no calcium, no magnesium (Thermo Fisher, Cat# 14190094, Waltham, MA, USA).FACS Max^TM^ Cell Dissociation Solution (Cat# T200100)RPMI 1640 medium (Gibco, Cat # 12633012).Fetal Bovine Serum (FBS), qualified, heat-inactivated (Gibco, Cat# 10500064, Waltham, MA, USA)Penicillin-streptomycin 100X (Gibco, Cat# 15070063, Waltham, MA, USA)Amphotericin b (Cat: PHR1662-500MG)EDTA 0.5 M (pH 8.0, without RNase) (Cat # AM9260G)Bovine Serum Albumin (BSA)
**Procedure**


Note: Prepare RPMI medium supplemented with 10% Fetal Bovine Serum (FBS), 3X Penicillin-Streptomycin, 12.5 µg/mL Amphotericin, and 2 mM EDTA. Additionally, prepare a solution of 1X DPBS supplemented with 1% BSA. Ensure that all reagents (excluding FACS MaxTM Cell Dissociation Solution) are stored at 4 °C one day before the dissection. It is imperative to maintain all the samples on ice.

Adults *Tg*(*mpeg1*:EGFP) fish with a size of approximately 3.6 cm (Figure 1A), should be euthanized in strict compliance with the guidelines outlined in the EU Directive 2010/63/EU on the protection of animals used for scientific purposes [5]. This could be achieved through an anesthetic overdose and rapid cooling. Specifically, immerse fish in 100 mL of E3 solution with 0.017% tricaine and place it in a recipient with ice. To ensure proper euthanasia, it is imperative to allow a five-minute interval following the final closure of the fish operculum.Using sterile forceps, transfer the fish, holding it by the tail, and immerse it subsequently in ice-cold 70% ethanol (for disinfection), cold sterile distilled water (to remove the ethanol), and finally, ice-cold 1X DPBS.Put the fish laterally in a cold (to favor tissue preservation) 94 mm diameter Petri dish beneath a stereomicroscope for further procedures (Figure 1B).Cut the epidermis and the underlying muscle from the gills to the anus with a sterile scalpel to expose the body cavity (Figure 1C). Using sterile forceps, remove the intestine (tissue with a tubular shape, semi-folded, and yellowish [6]) pulling from the anterior part.Remove all surrounding tissues (liver, gallbladder, swim bladder, and spleen) using sterile forceps. Rinse with ice-cold PBS 1X with a sterile Pasteur pipette to clean the whole intestine (Figure 1D).Immediately place the removed intestine in a 1.5ml microfuge tube with supplemented RPMI medium (Figure 1E). Keep it on ice while removing other intestines.With the help of sterile forceps, take the intestine and immerse it in 500 µL of FACS Max^TM^ Cell Dissociation Solution and incubate it for five minutes at room temperature. Subsequently, employ vortex and pipetting for an additional five minutes to facilitate tissue dissociation. **Note:** The cell suspension derived from one to four intestines can be filtered using the same cell strainer.Place a 70 µm cell strainer into a sterile 50 mL tube, keep it on ice, and carefully transfer the intestinal cell suspension onto the cell strainer using a 1 mL pipette.Proceed to rinse the cell strainer with an additional 1 mL of the FACS Max^TM^ Cell Dissociation Solution.Repeat step 8 using a 40 µm cell strainer, as illustrated in Figure 2A.Recover the entire cell suspension, including that on the bottom surface of the Cell Strainer (Figure 2B), place it into a microtube, and centrifugate it at 300× *g* for 5 min at 4 °C. During this step, an easily visible pellet will form, as seen in Figure 2C. Discard the supernatant and resuspend the pellet in 500 µL of DPBS supplemented with 1% BSA.Perform an additional centrifugation at 300× *g* for 5 min at 4 °C. Discard the supernatant and gently resuspend the pellet in 1 mL of the RPMI medium, enriched with 10% FBS, 3X Penicillin-Streptomycin, 12.5 µg/mL Amphotericin, and 2 mM EDTA. This step is essential for enhancing cell viability and preserving cell morphology.Position a 20 µm cell strainer within a sterile 5 mL round bottom polystyrene FACS tube and filter the cell suspension while maintaining the entire process on ice (Figure 2D).Maintain the cell suspension on ice until the FACS experiments are ready to be conducted (Figure 2E).Repeat the same procedure for the AB Wild-Type fish.

(2)
**
*Isolation of GFP^+^ subpopulation by FACS*
**
Notes: It is imperative to conduct FACS experiments promptly following tissue dissociation. If necessary, adjust the concentration of the cell suspension using the supplemented RPMI medium. We strongly recommend directly recovering the sorted cells into a lysis buffer to minimize RNA degradation [7,8].
**Materials**
Intestinal cells suspensionIce coolerRound Bottom Polystyrene FACS tube, 5 mL (Falcon, Cat#352052)RNase/DNase-free 1.5 mL microfuge tubes (Cat# CFT-001-015)P1000 pipettes and tipsMicrofuge tubes, 1.5 mL (Cat# CFT-001-015)
**Equipment**
BD FACSAria^TM^ Fusion Cell Sorter
**Reagents**
RPMI medium supplemented with 10% FBSi, 3X Pen-Strep and 12.5 μg/mL AmphotericinPropidium Iodide stain (Cat# P4864-10ML)RLT Buffer (lysis buffer provided by the RNeasy micro-Kit, Qiagen, Les Ulis, France)
**Procedure**


In our case, sorting procedures were conducted in a BD FACSAria^TM^ Fusion Cell Sorter, equipped with four spatially separated lasers (405 nm, 488 nm, 561 nm, and 633 nm). Calibration and a performance assessment of the instrument were carried out prior to the experiments.To prevent a decrease in the cell viability, preferably use a nozzle size of 100 µm for the whole procedure.Adjust the voltage settings as needed and set any compensation controls if necessary during the procedure.Incubate the intestinal cell suspensions with 1 mg/mL propidium iodide (PI) in a dilution of 1 µL PI in 1 mL of cell suspension (1:1000), to label all dead cells.We recommend employing a back-gating strategy to set up the sorting enrichment experiments.Identify the live GFP^+^ cells by gating the PI^−^ cells and GFP^+^ cells simultaneously in wild-type and *Tg*(*mpeg1*:EGFP) samples, using the following configuration: 695/40 band-passfilter and 488 nm excitation (PerCP-Cy5-5-A in the graph) to collet PI and 530/30 band-passfilter and 488 nm excitation (FITC-A in the graph) to collect GFP. Plot the graph in log mode (Figure 3A,B).Set up the cell autofluorescence using a dot plot in log mod of FITC-A vs. Pacific Blue-A (525/50 band-pass filter and 405 nm excitation) for the wild-type sample, establishing the limit of the green autofluorescence versus the blue autofluorescence simultaneously (Figure 3C,D).Separate the GFP^+^ cells from the *Tg*(*mpeg1*:EGFP) sample using the setting established in step 7, with the same detectors to ensure gating only GFP^+^ cells. We recommend establishing the gate usually between 10^2^ and 10^5^ in the FITC-A detector.We do not recommend discriminating cell doublets using the FSC parameter due to the high heterogenicity in size of these cells (Figure 3E,F).Identify the two GFP^+^ cell populations from the *Tg*(*mpeg1*:EGFP) sample by making a dot plot based on the GFP fluorescence and size (FITC-A vs. FSC-A), defining the sorting parameters for small GFP^+^ cells as FSC-A^low^ GFP^+^ cells and big GFP^+^ cells as FSC-A^high^ GFP^+^ cells (Figure 3G,H).Additionally, analyze the fluorescence peaks intensity of the small GFP^+^ and big GFP^+^ cells by histogram plots to uncover variations in GFP expression within each population, thereby facilitating the identification of distinct subpopulations (Figure 3I).Finally, sort the small GFP^+^ and big GFP^+^ cells, as well as GFP^−^ cells as a control.Ensure that the sorted cells are directly collected into a microfuge tube containing the lysis buffer and maintain them on ice.

(3)
*
**Total RNA Extraction Procedures**
*
Note: To ensure an RNA degradation-free environment, it is advisable to clean the workspace with an RNase decontaminating product and employ filter tips. Utilize the appropriate RNA extraction kit according to the number of cells recovered in each case and adhere to the manufacturer’s instructions meticulously. We recommend using the RNeasy Micro Kit—QIAGEN to perform a successful RNA extraction for a low number of cells (<5 × 10^5^ cells).
**Materials**
Sorted cellsRNeasy micro-Kit (Cat# 74104, Qiagen, Les Ulis, France)RNase/DNase-free 1.5 mL microfuge tubes (supplemented by RNeasy micro-Kit, Qiagen)P1000 pipettes and tips
**Equipment**
Cooling centrifugeVortex
**Reagents**
Ethanol AbsoluteUltrapure DNase/RNase-free distilled water (Cat# 10977035, Invitrogen, Boston, MA, USA)
**Procedure**


Prepare 70% and 80% ethanol solutions using Ultrapure distilled RNase-free water.Following FACS sorting, ensure that the collected cells in the lysis buffer are kept on ice or frozen at −20 °C.To ensure high-quality RNA, proceed with RNA extraction according to the Qiagen RNeasy micro-Kit procedure, following the manufacturer’s instructions.For optimal RNA concentration, elute the RNA in the column using 14 µL of RNase-free water and promptly place the RNA sample on ice.Set aside a minimum of 1 µL of RNA from each sample to assess its concentration and quality (Table 1).Store the RNA sample at −20 °C for short-term analysis or −80 °C for long-term analysis. However, it is preferable to perform cDNA synthesis as soon as possible.

(4)
*
**Analysis of the Gene Expression Levels in Zebrafish Intestinal Macrophages via RT-qPCR**
*

**Materials**
Total RNA from sorted cell populationscDNA from sorted cell populationsRNase/DNase-free 0.2 mL microfuge tubesPipettes and tips
**Equipment**
ThermocyclerReal-Time PCR System
**Reagents**
SensiFAST^TM^ cDNA Synthesis Kit (Meridian Bioscience, BIO-65053; Cincinnati, OH, USA)SensiFAST™ SYBR^®^ No-ROX Kit (Meridian Bioscience, BIO-98005, Cincinnati, OH, USA)Ultrapure DNase/RNase-free distilled water (Invitrogen, Cat# 10977035, Boston, MA, USA)Primer aliquots from each gene (*mpeg1.1, tnfα* and *il10*).
**Procedure**



*Note: This section comprises two parts. The first is the cDNA synthesis through retro-transcription, while the second encompasses the analysis of the gene expression levels in each of the intestinal macrophage populations.*


Carry out retro-transcription using the SensiFASTTM cDNA Synthesis Kit, adhering to the manufacturer’s recommended procedures.Following cDNA synthesis, conduct qPCR assays using primers aliquots from gene *mpeg1.1, tnfα* and *il10* (Table 2) and the SensiFAST™ SYBR^®^ No-ROX Kit.Perform data analysis using the QuantStudioTM Real-Time PCR System Software v1.3 and GraphPad Prism v9.0.1 software.

## 3. Results

GFP^+^ macrophages were isolated from the intestines of transgenic *Tg(mpeg1:GFP)* zebrafish using fluorescence-activated cell sorting (FACS). Intestines from wild-type AB line fish served as the negative controls. A back-gating strategy was developed to directly isolate GFP^+^ cells, with viable cells identified by propidium iodide (PI) incorporation, identified by FITC-A and PerCP-Cy5-A detectors, respectively (Figure 3A,B). A viability of 81.3% (P1) was achieved. To ensure the specific separation of GFP^+^ macrophages and to prevent signal leakage from other channels into the green channel, adjustments and corresponding offsets were applied in the FITC-A vs. Pacific Blue-A plots (Figure 3C,D). GFP^+^ events comprised 2.8% of total intestinal cells and 3.4% of viable cells (P2). Although doublets and multimers were assessed using forward scatter (FSC-A) and side scatter (FSC-H) parameters, no gating was applied due to significant heterogeneity in macrophage sizes (Figure 3E,F). We identified two GFP^+^ macrophage subpopulations based on cluster formation, indicating a noticeable size disparity in the cell subsets. To separate the cell subsets, we established a gate to separate small (P3) and large (P4) GFP^+^ cells, representing 7.5% and 23.2% of the total GFP^+^ cells, respectively (Figure 3G,H). Macrophages positive for GFP with an FSC-A parameter of less than 50 (×1000) were excluded from the analysis since they were considered cellular debris, as different voltage settings failed to separate this population from the *y*-axis, which is characteristic of cellular debris (Figure 3H). Going further, we relied on the fluorescence intensity of P3 and P4 populations represented in the histogram, revealing higher fluorescence intensity in large cells (P4) compared to small cells (P3) (Figure 3I). Once the gates were established, the GFP^+^ macrophages were sorted. From four intestines, we isolated a total of 20,931 small cells and 54,745 large macrophages. Then, we extracted the total RNA from each cell population using the QIAGEN RNeasy Micro Kit, specifically designed for RNA extraction from low cell numbers (<5 × 10^5^ cells). The concentrations of total RNA extracted were measured at 3.9 µg/µL and 9.2 µg/µL for small and large cells, respectively, with 260/280 absorbance ratios of 1.86 and 1.96 (Table 1). Gene expression levels in small and large macrophage populations, as well as non-macrophage GFP^−^ cells, were assessed via RT-qPCR. We focused our attention on *mpeg1.1*, a marker of all macrophages in zebrafish, *tnfα*, expressed by pro-inflammatory macrophages and *il10*, a marker characteristic of anti-inflammatory macrophages. We showed a high expression of the *mpeg1.1* gene in both small and large macrophage populations, contrasting with considerably lower expression in GFP-negative cells. Additionally, the *tnfα* gene exhibited higher expression levels, specifically in large cells compared to the small ones. Finally, *il10* gene showed significantly elevated expression in small cells compared to the large ones (Figure 4).

## 4. Discussion and Conclusions

In our study to identify macrophage subpopulations in non-traditional vertebrate models like zebrafish, we encountered different challenges, including the lack of specific molecular markers for their recognition. To address this fact, we adopted a strategy to isolate and characterize two intestinal macrophage subpopulations combining two properties, their GFP expression, and the difference in morphology and size observed in vivo. However, the generally small size of zebrafish cells makes it particularly challenging to distinguish whole cells from cellular debris [9]. For this reason, instead of the conventional approaches carried out in FACS experiments to separate groups of cells [9,10], we utilized a back-gating strategy in which we initially isolated cells based on GFP expression. This allowed us to skip the standardization of the basal separation between whole cells and cellular debris. The identification of GFP^+^ cells was simultaneously conducted through the recognition of viable cells by propidium iodide discrimination. This enabled us to exclude all dead cells that typically have high auto-fluorescence and can release intracellular material, causing intact cells to clump and contaminate sorted populations by releasing RNases, leading to a lower total RNA quality. Additionally, to ensure the specific separation of GFP^+^ cells and to prevent the filtration of signals from another channel towards the green channel, we realized adjustments and the corresponding compensations in the FITC-A (530/30 band-pass filter) vs. Pacific Blue-A (450/50 band-pass filter) dot plots. Finally, we were able to identify two well-defined cell groups with different sizes by plotting the GFP fluorescence vs. the morphological parameter FSC, which confirms the presence of intestinal macrophage populations segregated by size.

A further obstacle involved the total RNA extraction to analyze variations in gene expression within isolated macrophages. The low number of cells available makes it particularly difficult to obtain high-quality RNA [7,8]. To solve this issue, we emphasized increasing the viability of cells and preserving the morphology by including the use of supplemented culture medium RPMI to keep the cells [11] and a nozzle size of 100 µm with low pressure during the cell sorting experiments. On the other hand, it is relevant to highlight the use of an adequate RNA extraction kit considering the small number of sorted macrophages that can be obtained from zebrafish. Nevertheless, our results indicate that it is possible to detect different transcription levels of cytokines in the samples analyzed.

In summary, we introduce a strategy that efficiently segregates GFP^+^ macrophage populations based on both GFP expression and size. This separation enables subsequent high-quality RNA extraction, paving the way for a comprehensive analysis of the transcriptomic profile.

## Figures and Tables

**Figure 1 mps-07-00043-f001:**
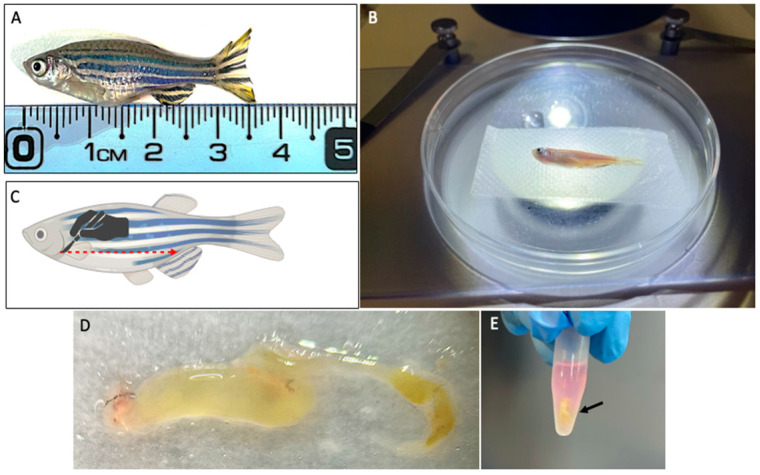
Intestine dissection from adult individuals. (**A**) An adult female in a lateral position, showing the size of the fish used. (**B**) The adult fish under the stereoscopic microscope in a 10 mm Petri dish before the intestine dissection. (**C**) Schematic representation of the lateral incision performed to dissect the intestine. (**D**) The whole intestine was extracted and cleaned from the surrounding tissue before disaggregation. (**E**) The intestine (black arrow) immersed in the RPMI medium immediately after dissection.

**Figure 2 mps-07-00043-f002:**
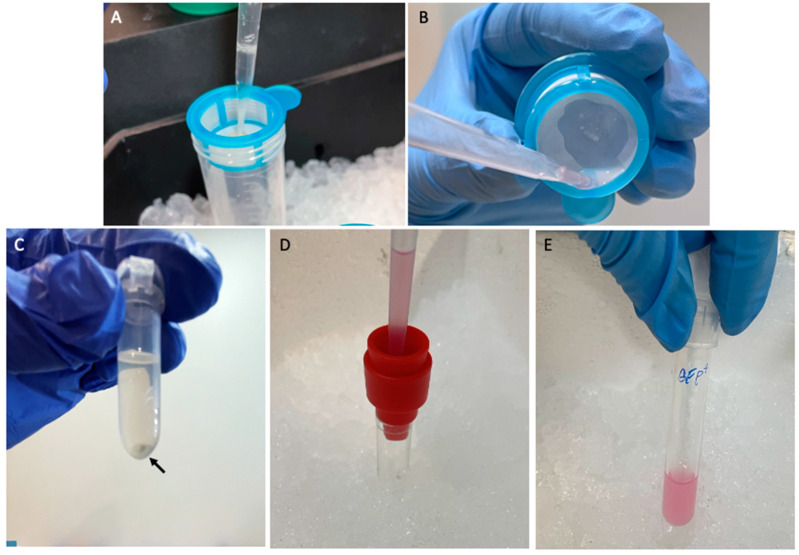
Obtention of the intestine cell suspension. (**A**) The intestinal tissue previously treated with FACs Max solution is filtered through a 70 µm and 40 µm cell strainer. (**B**) Recovery of the cell suspension at the bottom of the cell strainer after filtration. (**C**) Eye visible pellet (black arrow) obtained after 5 min of centrifugation at 300× *g*. (**D**) The pellet resuspended in 1 mL of RPMI medium supplemented with 5% FBSi, 3X Pen-Strep, 12.5 μg/mL Amphotericin and 2 mM EDTA is filtered through a 20 μm cell strainer. (**E**) Cell suspension is transferred to a 5 mL round bottom polystyrene FACS tube and kept on ice until separation.

**Figure 3 mps-07-00043-f003:**
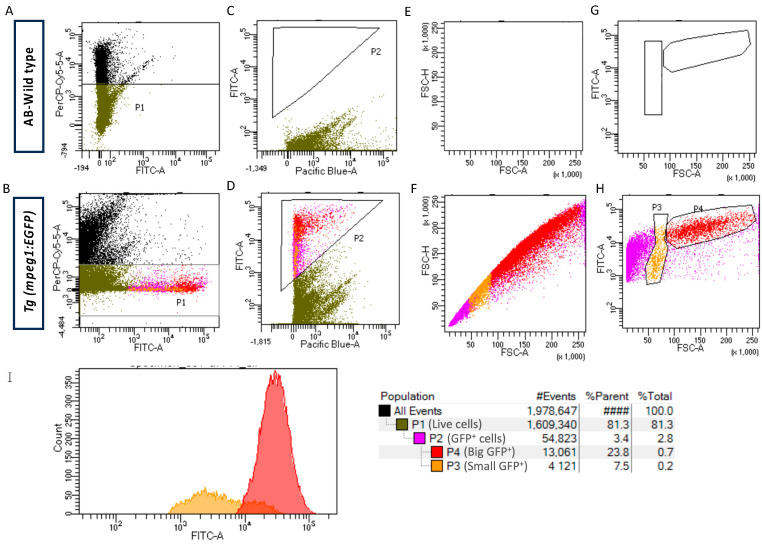
Isolation of morphologically different intestinal macrophages from *Tg (mpeg1:EGFP)* by FACS. (**A**,**B**) Representative FACS gating strategy of live GFP^+^ intestinal cells based on propidium iodide stain and green fluorescence in a PerCP-Cy5-5-A vs. FITC-A dot plot in log mode. (**C**,**D**) Representative gating for GFP^+^ cells using the FITC-A vs. Pacific Blue-A filters. (**E**,**F**) Representative dot plot of FSC-H vs. FSC-A of all live GFP^+^ cells. (**G**,**H**) Representative gating of two morphologically distinct subpopulations (GFP^+^ small size and GFP^+^ big size) is performed based on the FITC-A fluorescence vs. FSC-A. (**I**) A representative histogram plot illustrates the fluorescence in both the small GFP^+^ cells (depicted in yellow) and the large GFP^+^ cells (depicted in red). The plot displays a distinctive peak for each population, revealing a higher fluorescence intensity for the big GFP^+^ cells.

**Figure 4 mps-07-00043-f004:**
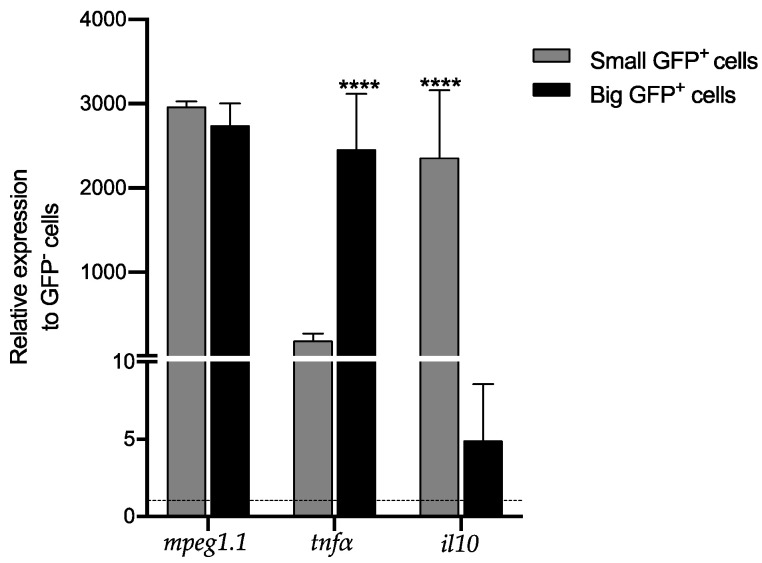
Pro- and anti-inflammatory cytokines are differentially expressed by small and big intestinal macrophages. The relative expression of the transcription levels of *mpeg1.1*, *tnfα*, and *il10* genes was analyzed in small GFP^+^, big GFP^+^ cells, and GFP^−^ cells. The graph shows the data normalized against the *ef1a* reference gene and compares the expression to those observed in GFP^−^ cells. **** *p* < 0.0001.

**Table 1 mps-07-00043-t001:** Representative number of GFP^+^ small and big intestinal macrophages sorted from adult *Tg(mpeg1:EGFP)* fish.

Population	No. of Intestines	No. of Cells	Total RNA Concentration	Ratio 260/280
Small GFP^+^ cells	4	20,931	3.9 ng/µL	1.86
Big GFP^+^ cells		54,745	9.2 ng/µL	1.96

**Table 2 mps-07-00043-t002:** Sequences of primers used in RT-qPCR.

Gen	Primer Sequence 5′-3′
*zEF1a.5*	TTCTGTTACCTGGCAAAGGG
*zEF1a.3*	TTCAGTTTGTCCAACACCCA
*zmpeg1.5*	GTGAAAGAGGGTTCTGTTACA
*zmpeg1.3*	GCCGTAATCAAGTACGAGTT
*zTNFa.54*	TTCACGCTCCATAAGACCCA
*zTNFa.34*	CCGTAGGATTCAGAAAAGCG
*zIL10.51*	CCAACGATGACTTGGAACCA
*zIL10.3*	CTAGATACTGCTCGATGTAC

## Data Availability

The data corresponding to the assays presented in this work are available at https://mega.nz/folder/PLQkTQqA#l5df9OOhA98ZTqCoV3sKCw.

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
