# Peer review of "Isolation of Intestinal Macrophage Subpopulations for High-Quality Total RNA Purification in Zebrafish"

_mps, 2024, doi:10.3390/mps7030043_

Round 1
Reviewer 1 Report
Comments and Suggestions for Authors
The characterization of intestinal macrophage subpopulations in zebrafish has not been well described due to the lack of specific molecular markers.
In the present work, Del Rìo-Jay and co-workers describe the development of a fluorescence-activated cell sorting strategy (FACS) for the insolation of different intestinal macrophage subpopulations in adult zebrafish taking advantage of the Tg (mpeg1: EGFP) transgenic line and the in vivo observation that macrophages undergo morphological changes in different microenvironments.
General comments:
The proposed method is clearly explained with accurate information regarding materials, equipment, reagents and a detailed technical step by step description of the procedures.
Specific comments:
Line 82: add the number [4] to the citation.
Line 91: please add information about the size and the characteristic of the forceps.
Line 109: delete the bullet point before Technologies.
Lines 130-132: add a brief description of what is made or prevented by this step.
Lines 135-137: add a reference for the intestine dissection or describe more in detail the procedure.
Line 147: Is it possible to process together two or more intestines? The same Cell Strainer can be reused for a second intestine or has to be discarded? Add a comment on that.
Description of Figure 1- Line 175: “immediately after dissection”.
Description of Figure 2- Line 179: in the caption is mentioned only the 40 microns Cell Strainer. To avoid misunderstanding cite also the 70 microns one.
Line 228: Figure 3 not Figure 4.
Line 238: There is no correspondence for Figure 3J.
Author Response
Many thanks for taking the time to review our manuscript. We took your comments and made the corresponding modifications. Please find the detailed point-by-point response below.
Line 82: add the number [4] to the citation.
Response: The citation number was added.
Line 91: please add information about the size and the characteristic of the forceps.
Response: We incorporated the following information: (Dumont #5; Cat# NC9889584)
Line 109: delete the bullet point before Technologies.
Response: It was corrected.
Lines 130-132: add a brief description of what is made or prevented by this step.
Response: We incorporated the following information: to favor tissue preservation
Lines 135-137: add a reference for the intestine dissection or describe more in detail the procedure.
Response: We reformulated the sentence as follows: Using sterile forceps, remove the intestine (tissue with a tubular shape, semi-folded and yellowish [6]) pulling from the anterior part. Reference 6 corresponds to this article: T. Gupta and M. C. Mullins, “Dissection of Organs from the Adult Zebrafish,” J Vis Exp, no. 37, p. 1717, 2010, doi: 10.3791/1717.
Line 147: Is it possible to process together two or more intestines? The same Cell Strainer can be reused for a second intestine or has to be discarded? Add a comment on that.
Response: We added the following Note: The cell suspension derived from one to four intestines can be filtered using the same cell strainer.
Description of Figure 1- Line 175: “Immediately after dissection”.
Response: We apologize for the mistake, we incorporated the word “after”
Description of Figure 2- Line 179: in the caption is mentioned only the 40 microns Cell Strainer. To avoid misunderstanding cite also the 70 microns one.
Response: We modified the sentence as follows: “… is filtered through a 70µm and 40µm cell strainer.”
Line 228: Figure 3 not Figure 4.
Response: We modified the sentence as follows: “using the setting sentence established in step 7”
Line 238: There is no correspondence for Figure 3J.
Response: We corrected the panel number, which is Figure 3I.

Reviewer 2 Report
Comments and Suggestions for Authors
The paper on the isolation of intestinal macrophage subpopulations for high-quality total RNA purification in zebrafish presents a meticulous approach and it seems a significant contribution to the described methodology.
Firstly, the clarity and coherence of the writing make the methods and results accessible to readers interested in specific techniques of zebrafish research and RNA purification. The authors present the experimental procedures and provide detailed explanations without overwhelming the reader with unnecessary technical vernacular.
I think one strength of this paper is its methodological rigor. The authors meticulously describe each step of the experimental procedures, including potential pitfalls and troubleshooting strategies, thus ensuring reproducibility.
In summary, this paper is well-written with clarity and rigor.
Minor points:
Fig 1 legend: where it reads: “immediately dissection” should be “immediately after dissection”.
Line 228. There is a reference for figure 4C that probably should be replaced with the information regarding the setting used.
The reference for Figure 3I-J in line 238 should not be before the reference to Figure 3 A to G.
Comments on the Quality of English Language
This paper is well-written with clarity and rigor.
Author Response
Many thanks for taking the time to review our manuscript. We took your comments and made the corresponding modifications. Please find the detailed point-by-point response below.
Fig 1 legend: where it reads: “immediately dissection” should be “immediately after dissection”.
Response: We apologize for the mistake, we incorporated the word “after”
Line 228. There is a reference for figure 4C that probably should be replaced with the information regarding the setting used.
Response: We modified the sentence as follows: “using the setting sentence established in step 7”
The reference for Figure 3I-J in line 238 should not be before the reference to Figure 3 A to G.
Response: We modified the order in which the panels of Figure 3 were cited.
